# Learning Compliant Box-in-Box Insertion through Haptic-Based Robotic Teleoperation

**DOI:** 10.3390/s23218721

**Published:** 2023-10-25

**Authors:** Sreekanth Kana, Juhi Gurnani, Vishal Ramanathan, Mohammad Zaidi Ariffin, Sri Harsha Turlapati, Domenico Campolo

**Affiliations:** School of Mechanical and Aerospace Engineering, Nanyang Technological University, Singapore 639798, Singapore; ksreekanth384@gmail.com (S.K.); juhigurnani@gmail.com (J.G.); vishal.pr@ntu.edu.sg (V.R.); mohd.zaidi@ntu.edu.sg (M.Z.A.); turl0001@e.ntu.edu.sg (S.H.T.)

**Keywords:** box-in-box insertion, compliant insertion, Learning from Demonstration, teleoperation, haptic feedback, human–robot collaboration, Gaussian mixture regression, barycentric interpolation, robotic automation

## Abstract

In modern logistics, the box-in-box insertion task is representative of a wide range of packaging applications, and automating compliant object insertion is difficult due to challenges in modelling the object deformation during insertion. Using Learning from Demonstration (LfD) paradigms, which are frequently used in robotics to facilitate skill transfer from humans to robots, can be one solution for complex tasks that are difficult to mathematically model. In order to automate the box-in-box insertion task for packaging applications, this study makes use of LfD techniques. The proposed framework has three phases. Firstly, a master–slave teleoperated robot system is used in the initial phase to haptically demonstrate the insertion task. Then, the learning phase involves identifying trends in the demonstrated trajectories using probabilistic methods, in this case, Gaussian Mixture Regression. In the third phase, the insertion task is generalised, and the robot adjusts to any object position using barycentric interpolation. This method is novel because it tackles tight insertion by taking advantage of the boxes’ natural compliance, making it possible to complete the task even with a position-controlled robot. To determine whether the strategy is generalisable and repeatable, experimental validation was carried out.

## 1. Introduction

Modern logistics require box-in-box insertion for a broad range of packaging tasks involving compliant, cardboard boxes. Industries such as electronics and cosmetics heavily rely on such box packaging. Typically, smaller boxes containing a product are stacked and placed into larger boxes for transportation. However, these tasks are often carried out manually on the factory floor, leading to drawbacks such as fatigue, limited operational time, and time consumption due to the repetitive and monotonous nature of the work. To overcome these challenges, there is a need for robot automation solutions to perform these tasks. Given that most industrial products are packaged in cuboid-shaped boxes, the problem at hand can be described as ‘developing an autonomous system that can perform compliant box-in-box insertion, wherein a cuboid box made of a flexible material is inserted into another larger box made of a similar deformable material’ (see Figure 1).

Nonetheless, autonomous insertion is a heavily researched field in robotics, especially in the classical peg-in-hole assembly problem [1]. The existing approaches fall into contact model-based and contact model-free approaches [2]. While the former covers techniques based on the contact state modelling between the parts, the latter focuses on learning paradigms such as Learning from Demonstration and Reinforcement Learning. Contact state modelling is known to be sensitive to uncertainties such as the elasticity of the system. A model-free strategy like Learning from Demonstration looks more promising given that the task in question requires compliant objects because humans’ superior sensory systems and decision-making skills enable them to accomplish insertion tasks even under uncertain environments.

While various research has attempted automated assembly in the past, the focus has frequently been on peg-in-hole insertion dealing with rigid objects because it is the basis of many assembly operations. To avoid the damage or wear of such objects, however, some compliance has been introduced through the use of force-guided robotic systems [3] combined with various other techniques, such as the use of machine learning [4] and vision [5], for example. Another simple approach used to safely manipulate objects with ease is to operate the manipulator in velocity control mode [6], which can then be translated to joint torques using the computed torque control method, as performed in [7], to introduce the desired compliance during manipulation. While several works take advantage of active interaction controls such as impedance/admittance control [8], compliance control [9], and hybrid force/position control [10], some other works also focus on utilising grasp planning [11] and soft grippers [12,13] to mimic the compliance of human fingers to reduce the effects of localisation uncertainties [7].

However, to the authors’ knowledge, there are very few works on a real-world industrial box-in-box application. More importantly, the existing works on the automation of box-in-box insertion tasks focus on the compliance of the robot itself or that of specifically designed soft grippers but not on the compliance of the objects themselves. This paper leverages the compliance of the objects along with Learning from Demonstration techniques to learn and automate a compliant box-in-box insertion task in the context of packaging an electric light bulb. The passive compliance introduced by the deformable cardboard boxes not only assists the operator in performing the tasks but also makes the autonomous insertion more tolerant of errors in localising the pose of the boxes.

The proposed approach is divided into three phases: the (a) demonstration phase, (b) learning phase, and (c) generalisation phase. In the demonstration phase, the user demonstrates the insertion task to the robot. The platform for teaching consists of two 7-DoF Kinova Gen3 Ultra-lightweight robot arms which form a master–slave teleoperated system whereby haptic feedback is always available to the user, as seen in Figure 2. While the active compliance provided by the master-robot enables the human to physically guides the robot effortlessly, the passive compliance introduced by the deformable cardboard boxes assists the operator in a smoother insertion by providing more lateral and angular tolerance. In the learning phase, the skill from human demonstrations is captured with probabilistic learning techniques, such as the Gaussian Mixture Model (GMM) and Gaussian Mixture Regression (GMR), which is then used for performing autonomous insertion. One key aspect of autonomous insertion is its capability to generalise for varying object poses. In the generalisation phase, a barycentric coordinate-based algorithm is implemented and evaluated to generalise the insertion for various object positions.

This paper is organised in the following manner. The related works on Learning from Demonstration and teleoperation are detailed in Section 2. The proposed framework is presented in Section 3 with an overview of the underpinning control system, the task description, and the details of the learning algorithms employed. The subsequent Section 4 comprises the experimental validation of this approach, followed by the conclusion and future work in Section 5.

## 2. Related Works

Learning from Demonstration (LfD) is a policy learning approach that involves a human demonstrator performing/demonstrating a task that is to be imitated by a robot [14]. This technique ends the need for a non-expert robot user to learn how to manually program for a task, as maybe required by other assembly task implementations such as in [15]. Learning from Demonstration can be implemented on robots in several ways, of which the most common ones are kinesthetic teaching [16], teleoperation [17,18], vision-sensing [19,20,21], and the use of wearable sensors. This paper focuses on a compliant packaging box-in-box insertion task using a slave robot arm that is controlled remotely by a master arm, in combination with vision-based sensing.

The datasets gathered from these demonstrations consist of task execution skills acquired from humans. These skills are then extricated (learnt) with the help of different techniques for the task to be performed by the robot [16]. One such commonly used technique is the Hidden Markov Model (HMM) [22,23], which is a robust probabilistic method to encode the spatial and temporal variabilities of human motion across various demonstrations [24] as a sequence of states. These states are defined as separate Gaussian Mixture Models (GMM) to explain the input data [25]. Ref. [23] compared the use of the HMM and Gaussian Mixture Regressions (GMR) approach vs. another, more popular Dynamic Movement Primitive (DMP) technique to allow robots to acquire skills through imitation. DMPs are basically units of action in terms of attractor dynamics of non-linear differential equations that encode a desired movement trajectory [26]. This method allows one to learn a control policy for a task from the demonstrations provided. However, standard DMP learning is prone to existing noise in human demonstrations [27].

It was also concluded by [23] that in the context of separated learning and reproduction, the HMM was more systematic in generalising motion than DMP. It allowed the demonstrator to provide partial demonstrations for a specific segment of the task instead of repeating the whole task again. This is an important feature when the aim is to refine one part of the movement.

Another study by [28], on the robotic assembly of mixed deformable and rigid objects, leveraged the use of haptic feedback with position and velocity controlled robots to make them compliant without explicitly engaging joint torque control. This information was integrated into a reinforcement learning (RL) framework as the insertion hole was smaller than the peg in diameter and was deformable. Thus, the contact mechanics for it were unknown, making it difficult to design a feedback control law. However, the lack of a vision system in their setup caused the insertion to sometimes fail. Another limitation of this method was that it only worked when the peg was relatively close to the hole. Research about automatic control helps with designing adaptive control [29,30] in cases where the peg position is different. Several other works utilising RL to adapt to varying dynamics of the environment during contact-rich robotic manipulation of non-rigid, deformable objects were reviewed by [31].

While the HMM and RL are widely used techniques for encoding task trajectories, these were not chosen for this paper due to the following shortcomings. RL requires a large dataset of demonstrations for training, which is an issue when there are time constraints, and the HMM method works by interpolating between discrete sets which are unsuitable for the continuous process of encoding tasks [32]. Also, the HMM relies on the proper choice of gains for stability, which requires estimating perturbations and the range of initial positions that the system can handle in advance, which can lead to inaccuracies [23]. Thus, in order to overcome these limitations, this work proposes a technique involving the use of GMMs for encoding training data and GMR for trajectory learning in combination with Barycentric coordinates for generalisation.

The GMM is a probabilistic method that is simple and robust enough for skill learning at the trajectory level. It enables the extraction of all trajectory constraints [16,21]. A combination of the GMM and GMR has been used to encode and generalise trajectories in previous independent works, such as [21,22,33]. The work by [22] presented a Learning by Demonstration framework for extracting different features of various manipulation tasks taught to a humanoid robot via kinesthetic teaching. The collected data were first projected onto a latent space for dimensionality reduction through the Principal Component Analysis (PCA) technique. Then, the signals were temporally aligned using the Dynamic Time Warping (DTW) approach for probabilistic encoding in GMMs. This study was able to successfully reproduce the important features of the performed tasks for different initial conditions. Ref. [21] made use of GMMs to encode upper-body gesture learning through the use of vision sensors and generalised the data through GMR.

While the GMM has been used to encode the training data for box-in-box insertion tasks, the proposed framework introduces a novel generalisation method based on GMR in Barycentric coordinates. Barycentric coordinates are commonly used in computer graphics and computational mechanics to represent a point inside a simplex as an affine combination of all of its vertices. This technique can be generalised to arbitrary polytopes [34] and is implemented in the generalisation approach of the training datasets. GMR is used to learn the robot trajectories that were demonstrated to the robot by the human operator and encoded by Gaussians. Regression makes it possible to retrieve a smooth signal analytically [22].

To perform human demonstrations of contact tasks like box-in-box assembly, it is important to consider how the task forces are being recorded and, more importantly, separated from the human forces acting on the system. Traditional kinesthetic teaching [35], i.e., by a human directly interacting with a robot and physically guiding it through the contact task, would result in the recording of the human forces coupled with the contact task forces. It is difficult to isolate the contact task forces only to perform force control later when generalising. One way to achieve this is to use two robots coupled through teleoperation control [36] so that the human interacts with the master robot and the slave robot deals with the contact task forces. Here, the term ‘Master-Slave’ simply means that the slave robot follows the motion of the master robot in the joint space. That is, to perform a task in a remote fashion, the human operator physically guides the master robot, thereby controlling the action of the slave robot which is in direct interaction with the environment. In fact, one of the first applications of teleoperation control was manipulation, specifically to handle radioactive materials from a safe distance in nuclear research [37].

For contact-rich tasks such as tight tolerance insertion, haptic feedback improves performance in the manipulation of remote objects [38]. Sensing in force-based teleoperation can be implemented through two possibilities: (a) a force/torque sensor attached to the flange of the robot to sense the end-effector forces and (b) a torque sensor built into every manipulator joint itself [39]. The latter approach was preferred due to the expensive nature of standalone six-axis F/T sensors. Instead, the robot construction was performed with strain gauges at every joint of an industrial robot manipulator [40]. Instead of using explicit force-control to track the contact task haptics, the teleoperation’s impedance characteristics are used to capture the forces that the slave should exert on the environment. This is inherent in the master kinematics and is a key aspect of the proposed approach, which allows for an open-loop playback of the assembly demonstrations given the task conditions do not change.

## 3. Methodology

### 3.1. Demonstration Phase

The goal in the demonstration phase is to successfully demonstrate the insertion of a folding carton (box B) containing an electric light bulb into the container (box A) shown in Figure 1. Kinesthetic teaching is one of the most widely used approaches in Learning from Demonstration paradigms for kinematic applications such as pick place and trajectory tracing. However, it is difficult to isolate the interaction forces purely due to insertion in the context of a contact task because robots with integrated joint torque sensors (such as the Kinova Gen3) also sense the forces due to human–robot physical interaction.

To allow the force feedback due to the contact task to be distinguished from the human forces, we choose to implement a master–slave teleoperated robot system using two Kinova Gen3 7-DoF robots. Continuous haptic feedback is provided to the human who interacts with the master side. The position information from the master side is used as the control input to the slave side. Consequently, all haptic feedback of the task on the slave side is fed back to the human operating the master side, as shown in Figure 2. As a result, the user is able to distinguish the interaction between geometric features such as the edges and faces of the cartons during the demonstration. The key advantage of this approach is the recording of the haptic feedback from a successful assembly task.

#### 3.1.1. Teleoperation Control

Two identical robots are teleoperated in a master–slave configuration to demonstrate the insertion task. The user controls the master, which in turn controls the slave to carry out the desired action. Figure 3 depicts a block diagram for a single robot arm. The detailed block at the top of Figure 3 depicts the robot dynamic compensation, while the condensed block below is a simplified version of the same, which will be further used to represent each robot arm in the system. The robot feedback for the 7-DoF robot arm used in this work comprises the (sensed) joint position q, joint velocitie $\dot{q}$, and joint torque $\tau \in R^{7}$, respectively. Each robot is calibrated [41] using an oracle to determine the robot parameters, i.e., joint offsets and link lengths, that will be used in the computation of robot parameters, including inertia M, Coriolis terms $C\in R^{7\times 7}$, and the gravity term $G\in R^{7}$, used in the standard robotic control Equation (Equation 1) later in this section.

For a master–slave system to be reliable, it is important that the master offers minimal resistance to the user for smooth guidance. Therefore, we define the torque $\tau_{dyn}$ (see Figure 3) originating from the dynamic effects (such as Inertia, Coriolis, and Gravity) that must be compensated during motion. Dynamic compensation is also crucial on the slave side, as the slave’s motion is to be solely dictated (ideally) by an external command torque ($\tau_{cmd}$) which will be a function of the master’s state variable, as defined later on.

Therefore, with reference to Figure 3, the resultant robot dynamics can be written as:(1)$$M\left(q\right)\ddot{q}+C\left(q,\dot{q}\right)\dot{q}+G\left(q\right)=\tau_{ctrl}$$
where $\tau_{ctrl}=\tau_{dyn}+\tau_{cmd}$ is the control torque issued to the robot, which should comprise (i) the dynamic compensation $\tau_{dyn}$, i.e., rendering the robot free to move in space, and (ii) external command torque $\tau_{cmd}$.

The block at the bottom of Figure 3 is a simple representation of the control block of any robot *i*, with the associated control and feedback signals marked by the superscript ${\left(\cdot \right)}^{i}$. In order to establish a master–slave behaviour, two such robots are connected together as depicted in the block diagram in Figure 4. As per the notation in the control block diagram above, we mark the control and feedback signals for master and slave with superscripts *m* and *s*, respectively (for example, the feedback joint configuration for the slave is represented as $q^{s}$).

Here, the external command torque to the slave is elastic in nature and is proportional to the joint position (vector) differences between the master and the slave. That is,
(2)$$\tau_{cmd}^{s}=K\left(q^{m}-q^{s}\right)$$
where $K\in R^{7\times 7}$ is the stiffness matrix. A torque-controlled slave robot obeying the above control law always follows the master, which is physically guided by a user. Through proper modulation of the stiffness elements in $K=\mathrm{diag}(\left[K_{1}\;K_{2}\ldots K_{7}\right])$, we programme the compliant behaviour of the slave robot, limiting the forces of interaction with its environment to mitigate the damage in the event of a collision.

For a master–slave system to be reliable, it is important that the master offers minimal resistance to the user for greater haptic transparency, i.e., rendering (as closely as possible) of slave-side contact forces. In addition to the dynamic compensation being conducted on both robots in the teleoperation scheme, we impose positive feedback on the master side to overcome the damping in the joints.

To that end, the external command torque on the master side is defined as follows:(3)$$\tau_{cmd}^{m}=\tau_{int}^{s}+B_{0}{\dot{q}}^{m}$$
where $\tau_{int}^{s}=\tau^{s}-\tau_{dyn}^{s}$ is the interaction slave torque (i.e., the torque due to physical contact on the slave robot; the reader is referred to Figure 3), that results from the difference between the sensed feedback torque and the dynamic torques for the slave robot arm. $B_{0}\in R^{7\times 7}$ is the diagonal damping matrix for the master side, and it was regressed from the experimental data to compensate for the joint damping torques, i.e., $B_{0}{\dot{q}}^{m}$. The structure for this damping term is given by a diagonal matrix with each value corresponding to the joint damping $B_{0}=\mathrm{diag}\left(\left[B_{1}\;B_{2}\ldots B_{7}\right]\right)$.

#### 3.1.2. Task Demonstration

With the teleoperated robot system devised above, the user demonstrates the 4 stages of the task, which are the (a) picking, (b) placing, (c) insertion, and (d) retraction of the robot. In the demonstration, a set of four distinct locations is considered for box B in the task space, while keeping box A at a pre-defined position. The combination of all the sub-tasks corresponding to every Box B location is referred to as an experiment, and for each experiment multiple trials are performed for all the four sub-tasks. The higher the number of demonstrations, the more reliable the encoding of the human skill, as probabilistic approaches such as GMM and GMR tend to identify the most common trends in the demonstrations. Over the course of the demonstrations, the spatial vector containing the robot Cartesian poses $d_{s}=\left[x,y,z,\alpha ,\beta ,\gamma \right]\in R^{6\times 1}$ (where *x*, *y*, and *z* are the Cartesian position coordinates and α,β, and γ are the Euler angles about *X*, *Y*, and *Z* axes, respectively) is to be collected separately for each of the sub-tasks indexed against the timestamp data $d_{t}\in R$. The collected data from multiple trials are then used to learn how each sub-task is performed by the human.

### 3.2. Learning Phase

In the learning phase, we use GMM and GMR to obtain the statistical average across multiple trials in the demonstration, as discussed in this section.

#### 3.2.1. K-Means Clustering

To fit a Gaussian Mixture Model to a dataset, an initial guess for the mean and the covariance of the Gaussians is required. For this purpose, K-means clustering is used on the dataset. Note that the learning is performed for each sub-task (with a specific initial location of box B and the destination of box A) separately so that the learnt individual sub-tasks can be reused separately in the future.

The data for the robot Cartesian pose (for a sub-task) for a total of *N* observations can be expressed as:(4)$$D=[d_{1},d_{2},\ldots ,d_{N}]$$
where each observation (also called the data point) $d_{j}={\left[d_{t},d_{s}\right]}^{T}\in R^{n}$ is indexed against the timestamp $d_{t}\in R$ and a spatial component $d_{s}\in R^{\left(n-1\right)}$, where *n* is the dimension of the data point. In this work, n=7, because the Cartesian pose is described by 6 kinematic variables.

The parameters for the Gaussian Mixture Model (GMM) are initialised by performing the K-means clustering algorithm.
(5)$$\left[\pi_{ini},\mu_{ini},\Sigma_{ini}\right]=\mathrm{Kmeans}\;\left(D,K\right)$$
where *K* is the number of clusters.

Here, $\mu_{ini}=\left[\mu_{1},\ldots ,\mu_{K}\right]$, with initial mean $\mu_{j}\in R^{n}$ and $\Sigma_{ini}=\left[\Sigma ,\ldots ,\Sigma_{K}\right]$ and with initial covariance $\Sigma_{j}\in R^{n\times n}$ for the *K* Gaussians with probability densities $N\left(\mu_{1},\Sigma_{1}\right),N\left(\mu_{2},\Sigma_{2}\right),\ldots ,N\left(\mu_{K},\Sigma_{K}\right)$, and $N_{j}\in R^{n}\times {Sym}^{+}\left(n\right)$ (${Sym}^{+}\left(j\right)$ denotes the manifold of j×j Symmetric Positive-Definite (SPD) matrices. Finally, we have $\pi =[\pi_{1},\pi_{2},\ldots ,\pi_{k}]$, $\pi_{j}\in R$ representing the prior probabilities for each of the Gaussian components.

#### 3.2.2. Expectation Maximisation (EM)

The process of Expectation Maximisation [42] estimates the parameters for the Gaussian Mixture Model by performing a recursive 2-stage (the E-step and the M-step) computation until convergence. The expectation (E-step) computes the contribution (known as the weight) of each of the Gaussians in the occurrence of each of the data points, while maximisation (the M-step) updates the initial guess for the Gaussian parameters based on the computed weights. Hence,
(6)$$\left[\pi ,\mu ,\Sigma \right]=\mathrm{EM}\left(D,\pi_{ini},\mu_{ini},\Sigma_{ini}\right)$$
where $\pi \in R^{n},\mu \in R^{n\times K}$ and $\Sigma \in R^{n\times n\times K}$ are the updated parameters (Refer to Appendix A).

As a result, a Gaussian Mixture Model (GMM) with *K* components can be used to model the dataset, with the following probability density function
(7)$$p\left(d_{j}\right)=\sum_{j=1}^{K}\pi_{j}N\left(d_{j};\mu_{j},\Sigma_{j}\right)$$
where
(8)$$\mu_{j}=\left[\mu_{t,j},\mu_{s,j}\right],\;\Sigma_{j}=\left[\begin{array}{cc} \Sigma_{tt,j} & \Sigma_{ts,j}\\ \Sigma_{st,j} & \Sigma_{ss,j} \end{array}\right]$$
and $\mu_{t,j}\in R$, $\mu_{s,j}\in R^{n-1}$, $\Sigma_{tt,j}\in R$, $\Sigma_{ts,j}\in R^{n-1}$, and $\Sigma_{ss,j}\in R^{\left(n-1)\times (n-1\right)}$.

#### 3.2.3. Gaussian Mixture Regression (GMR)

Once GMM is fitted to the obtained dataset, the goal is to extract a single average trajectory through GMR; the trajectory is characterised by all the demonstrations but weighted by the probability of their occurrence. In addition, GMR provides a smooth trajectory despite any discontinuous or jerky motions present during the demonstration.

A regression of the obtained Gaussian mixture model over a temporal vector $T=[t_{1},t_{2},\ldots ,t_{m}]$ can be performed as [22]
(9)$$\left[\mu_{r},\Sigma_{r}\right]=\mathrm{GMR}\left(\pi ,\mu ,\Sigma ,T\right)$$
where $\mu_{r}\in R^{(n-1)\times m}$ and $\Sigma_{r}\in R^{(n-1)\times (n-1)\times m}$ are the conditional expectation of the data and associated covariances corresponding to the timestamp values in T (refer to Appendix B).

Performing GMR outputs a trajectory $r\left(t\right)=[\left(t_{1},\mu_{1}\right),\left(t_{2},\mu_{2}\right),\ldots ,\left(t_{m},\mu_{m}\right)]$ for each sub-task, and the overall robot trajectory for the $k^{th}$ experiment is characterised by executing the trajectories for all the subtasks, namely pick, place, insert, and retract ($r_{p_{k}}\left(t\right),r_{pl_{k}}\left(t\right)$, $r_{in_{k}}\left(t\right)$, and $r_{re_{k}}\left(t\right)$, respectively) in order.

### 3.3. Generalisation Phase

The demonstrated trials, though performed for a selected set of object poses, can be generalised to arbitrary object positions. Barycentric coordinates, a widely used notion in computer graphics, especially in texture mapping and ray tracing applications, can be utilised for this.

**Note:** This work does not address the problem of generalisation for varying object orientations.

#### Trajectory Generalisation Using Barycentric Coordinates

For a 2D convex polygon *P* with *n* vertices $V=[v_{1},v_{2},\ldots ,v_{n}]$, one can define a set of barycentric coordinates $\phi =\left[\phi_{1},\phi_{2},\ldots ,\phi_{n}\right]:P\to R^{n}$ such that an arbitrary object position $x\in R^{2}$ within the convex hull of *P* can be expressed [34] as
(10)$$\sum_{i=1}^{n}\phi_{i}\left(x\right)v_{i}=x\;\forall \;x\in P\;$$
from the linear reproduction property wherein
(11)$$\sum_{i=1}^{n}\phi_{i}\left(x\right)=1\;\forall \;x\in P\;$$
from the partition of unity property.

Bringing this to the task perspective (See Figure 5), assuming that the initial object positions in the 2D plane, which are observed by the robot at time $t_{p}$ (in the picking phase of the trial) across *m* trials, i.e., $p=[p_{1}\left(t_{p}\right),p_{2}\left(t_{p}\right),\ldots ,p_{m}\left(t_{p}\right)$]: $p_{i}\left(t_{p}\right)\in R^{2}$, form a convex polygon, for *m* different initial positions of box B, any arbitrary object position $p_{0}$ (for brevity, henceforth, we omit indicating that the barycentric coordinates are being evaluated on the demonstration data at time $t_{p}$, i.e., for different initial positions of box B at the time of picking) within the polygon can be represented in terms of a set of barycentric coordinates $w_{1},\ldots w_{m}$ such that:(12)$$p_{0}=\sum_{i=1}^{m}w_{i}\;p_{i}$$

To ensure that $\sum_{i=1}^{m}w_{i}=1$, out of all the possible sets of barycentric coordinates, the considered affine functions possessing the minimal l-2 norm [43] are computed as:(13)$$w_{i}=\langle p_{0}-c,{\left(VV^{T}\right)}^{-1}v_{i}\rangle +\frac{1}{m}$$
where 〈·,·〉 is the inner product and $V=\left[v_{1},\ldots v_{m}\right]\in R^{2\times m}$ for each of the pick-phase object positions from demonstrations, with $v_{i}=p_{i}-c$ and $c=\frac{1}{m}\sum_{i=1}^{m}\;p_{i}$.

In this work, to ensure the geometric nature of the generalisation capability to pick the box from a new position, we consider the use of barycentric coordinates. This is guided by the intuition that if the box is placed in the exact same position as in the original demonstration, the reproduction trajectory should be no different from the demonstration. Conventional generalisation techniques such as TP-GMM compute the reproduction trajectories by way of products of task-parameterised Gaussians (using a quadratic error cost, i.e., Equation (Equation 3) [44] to prioritise the frame that contributed more to the product Gaussian’s mean at a certain time). For this reason, TP-GMM would not generate an exactly identical trajectory given an exactly identical object position. Instead, the reproduction trajectory will be a weighted average, with the weights chosen based on probabilities and not geometry. Thus, it is a probabilistic approach that allows one to prioritise which task frame was important at different times of the task [45] based on the variance in the demonstration as observed from that frame. For this reason the TP-GMM formulation works best when the number of initial poses shown are maximised, since their contribution is weighted in the computation of the task-parameterised variance. Therefore, the reproduction trajectory will not be identical to the original demonstration even if the new object pose is identical. Using the geometric nature of barycentric coordinates, we overcome this challenge. Moreover, the use of barycentric coordinates may be deployed in rotations as well, since this method is geometrically invariant (i.e., the coordinates $w_{i}$ may be applied to any vector-space extrapolation, not necessarily in the Euclidean space of position vectors $p_{i}$), allowing one to expand the generalisation capability to rotation-only generalisation, e.g., when the box is at the same position but just rotated, making the use of barycentric coordinates a novel contribution to generalisation capabilities in LfD scenarios.

## 4. Experimental Validation

In this section, the proposed framework is validated by performing an autonomous box-in-box insertion test followed by a repeatability test with arbitrary box B positions. Note that these experiments focus on how well this framework performs for varying object positions while considering the orientation to be constant. The master–slave setup implemented in this paper involves two 7-DoF Kinova Gen3 ultra-lightweight robotic arms. The master robot arm is mounted with a customised 3D printed handle to provide the user with the convenience of guiding, whereas the slave is equipped with a 2F-85 Robotiq gripper for object manipulation. Both the robots are controlled in torque control mode from a single workstation at a frequency of 1 kHz through TCP/IP. Before the task demonstration was performed, the procedure for fast kinematic re-calibration of cobots [41] was performed on the slave robot to account for any joint offsets that need to be regressed. The results from this experiment are listed in Table 1. Additionally, the gravity compensation module on both the robots was tested to verify there is no drift in the robots. In case of drift, the joint torque offsets of the robots were reset after initialising the robot to the zero configuration. This is an important step to ensure the human does not feel unnecessary forces originating from torque sensor offsets fed back because of the torque controller, i.e., the intended function is so that the human feels the haptic feedback from the task only. Once the robots are initialised with these steps, the demonstration of the insertion task of the box-in-box assembly is carried out.

sensors-23-08721-t001_Table 1Table 1Experimental parameters and list of acronyms.Quantity NameSymbolValue and UnitsStiffness for joint *i*

Ki

15 Nm/radJoint 1 damping for master

B1

3.3132 NsJoint 2 damping for master

B2

3.4284 NsJoint 3 damping for master

B3

3.1245 NsJoint 4 damping for master

B4

3.2895 NsJoint 5 damping for master

B5

2.836 NsJoint 6 damping for master

B6

2.8579 NsJoint 7 damping for master

B7

7.6092 NsPick time

tp

≈11 sPlace time

tpl

≈25 sInsert time

tin

≈34 sRetract time

tre

≈40 s
**Full form**

**Acronym**
Learning from DemonstrationLfDGaussian Mixture ModelGMMGaussian Mixture RegressionGMRExpectation MaximisationEMInternet Protocol SuiteTCP/IP

### 4.1. Task Demonstration

The overall insertion task was divided into four sub-tasks: (see Figure 6). Here, the geometry of box B is that of a square prism and is placed in such a way that it rests on one of its bases.

In the picking sub-task, both the master and the slave robots initially assume a pre-defined home position. The teleoperated setup enables the user to control the slave motion by guiding the master robot. The user guides the master in such a way that the slave approaches box B with its gripper oriented at a suitable angle for grasping. As soon as the robot is ready to pick, the gripper is closed, ensuring a firm grasp.

Once box B is picked up, the user enters into the insertion sub-task, where the user guides the slave towards box A. The placing and insertion sub-tasks are interlinked, as the mechanism of insertion determines the way box B is placed at the opening of box A.

Considering that the task at hand is a tight-tolerance insertion task, insertions performed with a vertical approach vector are highly likely to fail. As such, the passive compliance provided by the boxes is exploited to perform a successful insertion. This can be seen in the insertion sub-task, divided into three sub-stages, with reference to Figure 7.

At the end of the placing sub-task, box B is oriented in such a way that one of its vertices is placed around the centre of the opening of box A.

Subsequently, the vertex of box B is pushed against the edge, followed by a simultaneous rotation about the X,Y axes and a translation along the *Z* axis. Here, the compliance introduced by the elastic nature of box B guides box A into B, assisting with the insertion task. Once the insertion is performed, the gripper is opened and the retraction sub-task is executed to move the robot away to a safe position.

In this demonstration, a set of distinct locations is considered for box B in the task space, while keeping box A at a pre-defined position. For each experiment, a new box location is considered, and for each box location, multiple demonstrations are performed (referred to as trial) for all the four sub-tasks. The higher the number of demonstrations, the more reliable the encoding of the human skill, as probabilistic approaches such as the GMM and GMR tend to identify the most common trends in the demonstrations. In addition, humans learn and adapt to a task (such as insertion) after repeated trials, identifying a successful strategy of action. Over the course of the demonstrations, the robot Cartesian pose data [x,y,z,α,β,γ] (where x,y, and *z* are the Cartesian position coordinates and α,β, and γ are the Euler angles about the X,Y, and *Z* axes, respectively) are logged separately for each of the sub-tasks along with the temporal data *t* at a frequency of 1 kHz (refer to Figure 8).

### 4.2. Learning Autonomous Insertion with GMM and GMR

The resulting Gaussian Mixture Regression is carried out for robot-pose data and is depicted in Figure 8 with the number of clusters, K=7, for the fitting of the Gaussian Mixture Model.

The fitted Gaussian Mixture Regression is then used to determine whether the slave robot (in position control mode) can successfully insert arbitrary object (box B, in our case) positions. An insertion is marked as successful if and only if box B is fully inserted in box A causing no damage to either of the boxes. In the event of a partial insertion, no insertion, or insertion causing damage to the boxes, the attempt is considered unsuccessful. Throughout this experiment, box A assumes the same position as it did in the demonstration phase. The various sub-tasks performed autonomously by the robot for a single successful trial are illustrated in Figure 9.

### 4.3. Generalisation and Repeatability Test

Six trials of autonomous insertion were carried out, with box B assuming six different positions within the convex polygon formed by the demonstrated positions. For each trial, the built-in RGB and depth camera on the Kinova Gen3 were used to identify the position of the box, which was followed by the barycentric coordinate-based trajectory adaptation. It was observed that the robot performed successful insertion for all six trials.

The repeatability of insertion was tested out by performing 10 trials corresponding to one of the box positions. The object was always oriented in such a way that its edges were in parallel with the base frame *X*-*Y* axes. Rather than just performing a simple repeatability test, it was important to analyse how well this framework performs under slight variations in the camera-identified object pose. For instance, if the object is manually reinstated back to its original pose after each trial, there exist slight deviations from the initially identified pose (by the camera). To test how this affects the insertion task, the robot assumes that the object pose is the same as the one identified in the first trial, where, in fact, slight variations were introduced in successive trials due to the manual placement.

In all 10 of the trials, the robot (operating in position control mode), successfully performed an insertion (for the video depicting the repeatability tests for two different box positions inside the polygon: video: refer to Appendix A). Even though the success of insertion highly depends on the repeatability of the robot itself, the results show that despite the slight variations in the object position and orientation arising from the manual placement (which is not captured through vision), the robot is able to perform the insertion successfully. The passive pose adjustment of the object when the gripper closes and the compliance provided by the boxes during the insertion allows such a position-controlled robot to accomplish these insertions successfully.

### 4.4. Discussion

Beyond the trivial advantages of having ease of programming with quick human demonstrations, our choice in using the GMM as the learning method allows for fast training compared with deep neural networks. This is because of the low dimensionality of the GMM, i.e., in the parameters used to train the model. From a practitioner’s perspective, no more than 10 demonstrations, each less than a minute long, were sufficient to automate the task of box-in-box insertion shown in this work.

The choice of the GMM also makes the model more explainable. Specifically, with time as one of the parameters, the choice of the number of Gaussian kernels directly relates to the number of “phases” in the experiment. For instance, a practitioner may directly choose this parameter based on their experience of the task. This contrasts with deep neural networks, where choosing the number of parameters/layers/activation functions is an open problem. The generalisation in this work was conducted using barycentric coordinates, which allow one to generate robot motion for a new object pose given visual feedback. However, this generalisation does not account for any adaptation needed in case of a task failure. This is one of the future threads to explore.

The importance of compliance has been studied in assembly tasks from the early 1980s [46] and, more recently, in [47]. Typically, lateral and angular compliances are required for at least one of the mating parts, to account for the errors encountered due to various sources of uncertainties in the task. The concept of a remote centre of compliance allows for free rotation/translation to reduce resistance and may be implemented using passive mechanisms [46], active compliance control [48], and by learning compliant motion primitives [49] from demonstrations that encode the correlation between compliance and tolerance required in the task. In this work, we rely on the passive compliance inherent in the paper boxes used for assembly instead of allowing for tolerance errors during insertion.

## 5. Conclusions and Future Work

Autonomous insertion tasks hold a great deal of significance in industrial packaging applications. This work presented a box-in-box insertion framework that uses the Learning from Demonstration paradigm in contrast to the conventional programming-intensive approaches. To respect the contact-rich nature of the insertion task, a master–slave teleoperated robot system was developed where the forces of interaction with the environment are effectively perceived by the user through continuous haptic feedback. The human skill from these demonstrations is captured with the help of the Gaussian Mixture Model and Gaussian Mixture Regression. The learned skill can be applied to different positions (of the box to be inserted) in the task space and is only constrained by the boundaries of a polygon formed by the demonstrated object positions, as shown by the experimental analysis. In the repeatability test conducted, it was observed that despite the presence of slight variations in the estimated object pose, the robot was able to successfully perform the insertion.

The proposed approach mainly relies on the compliance offered by both the folding cartons, which facilitated smooth guidance during the insertion sub-task. In other words, compliance is provided by the environment rather than the robot, which enables successful insertion. This work only focused on how the demonstrated skill can be adapted to varying object positions. Our future work will incorporate the variation in the object orientation as well into the framework. Additionally, impedance control can potentially be implemented on the robot to achieve minimal contact forces in the event of an unsuccessful insertion.

## Figures and Tables

**Figure 1 sensors-23-08721-f001:**
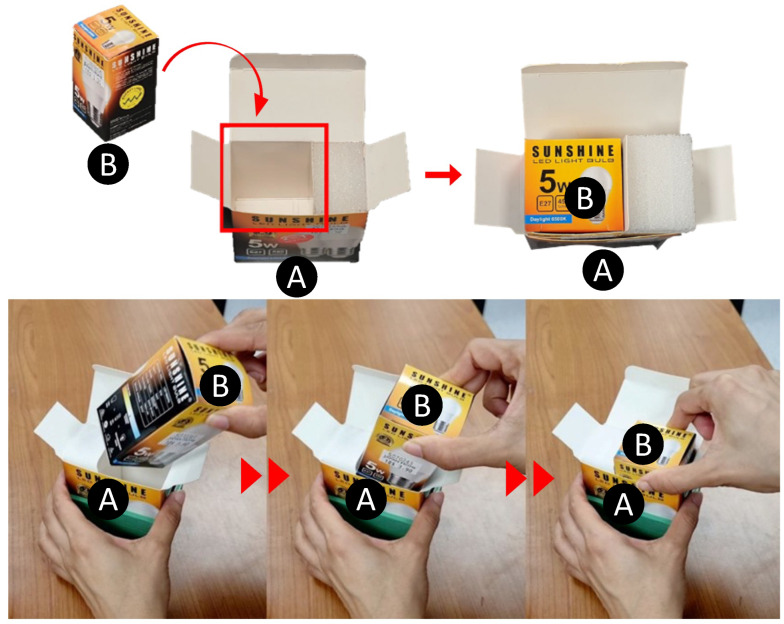
Example of a compliant box-in-box insertion where a folding carton (box B) containing an electric light bulb is to be inserted into a receptacle folding carton (box A). Manual insertion performed by a human (**bottom**).

**Figure 2 sensors-23-08721-f002:**
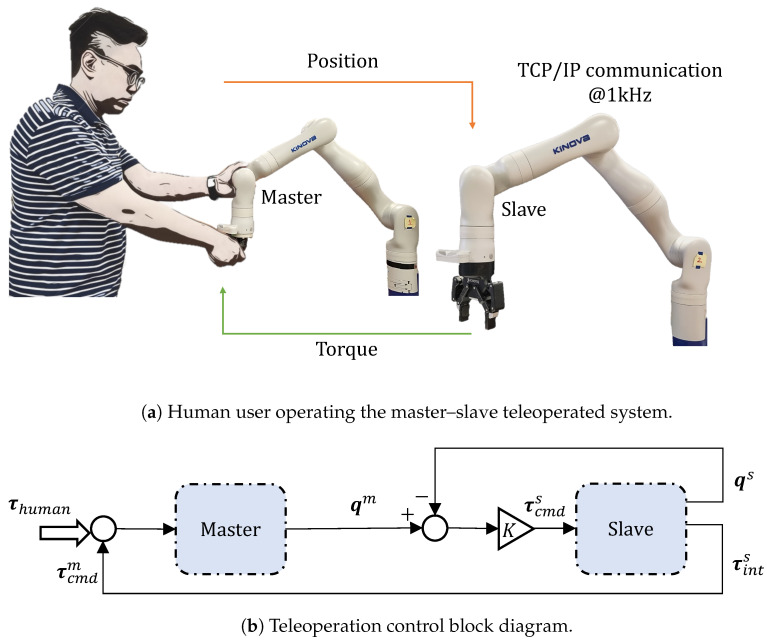
Teleoperated system of two 7-DoF Kinova Gen3 robots.

**Figure 3 sensors-23-08721-f003:**
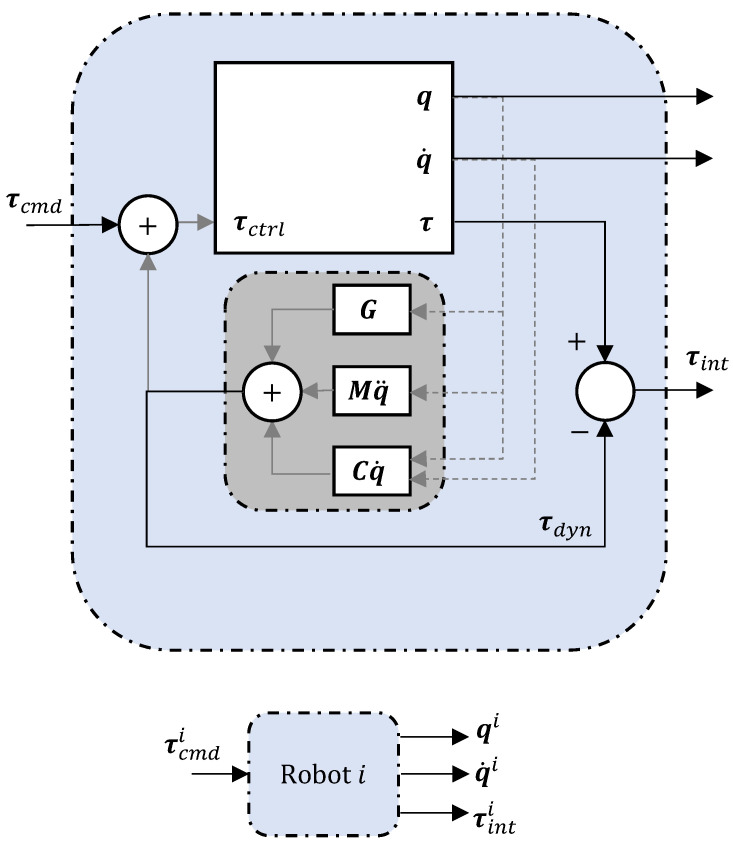
Torque controlled robot with dynamic compensation.

**Figure 4 sensors-23-08721-f004:**
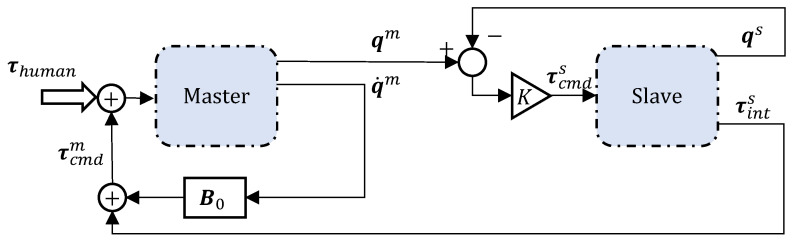
Master–slave control diagram for teleoperation with physical interaction with humans, captured by $\tau_{human}$.

**Figure 5 sensors-23-08721-f005:**
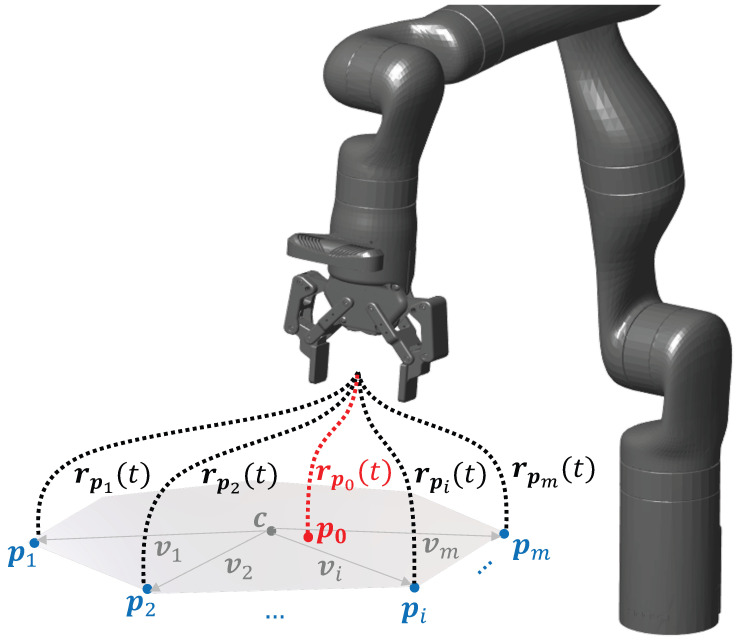
For an arbitrary object pick-position $p_{0}$ within the convex hull of the polygon formed by the demonstrated positions, the trajectory of the robot (red) can be represented as an affine combination of the demonstrated trajectories (blue).

**Figure 6 sensors-23-08721-f006:**
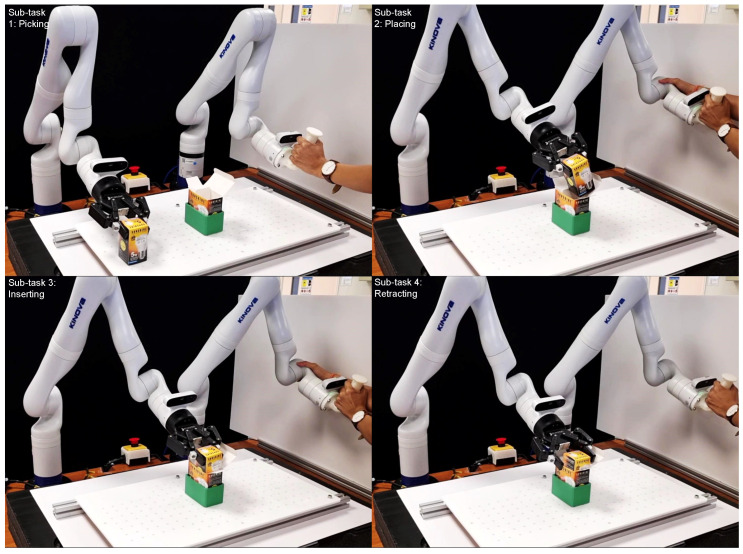
Demonstration of pick and insert task performed using master–slave teleoperation between two KINOVA Gen3 robotic arms. User moves the master robot to guide the slave robot to perform the following sequence of tasks: (video: refer to Appendix A).

**Figure 7 sensors-23-08721-f007:**
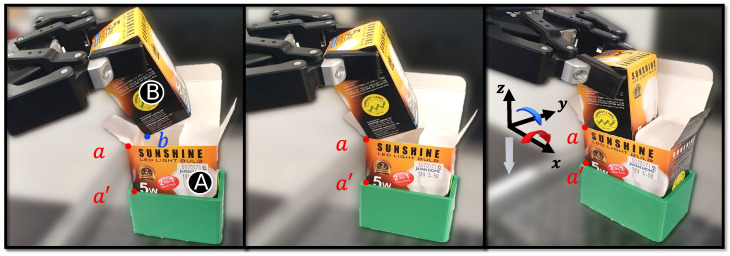
The insertion sub-task is performed in 3 sub-stages. In the first stage, box B is oriented such that one of the corners is placed around the centre of box A. This is followed by aligning the corners of the boxes, followed by the final insertion.

**Figure 8 sensors-23-08721-f008:**
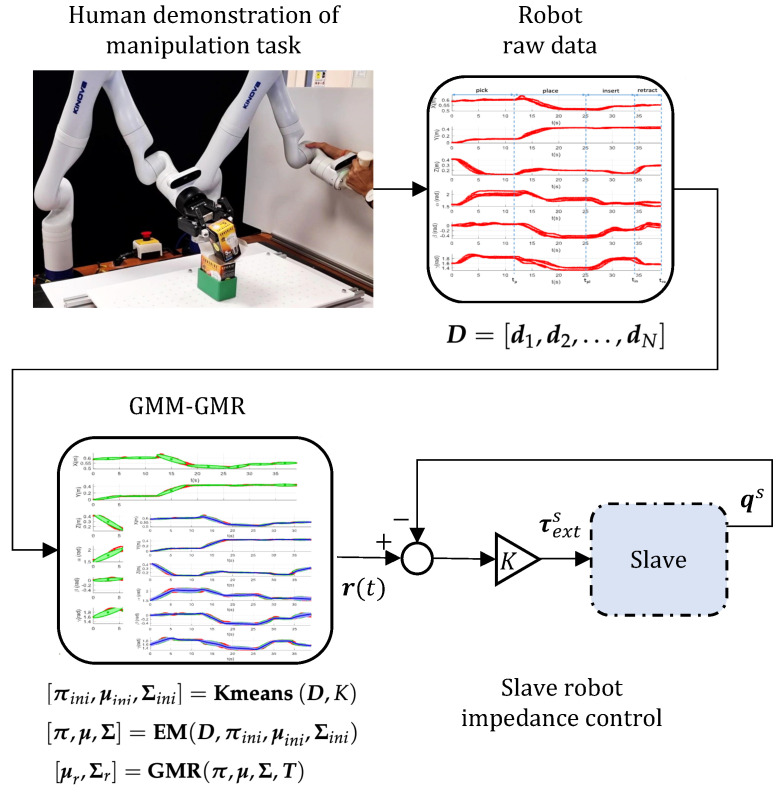
From the human demonstration of manipulation task, sample dataset with $d_{i}={\left[t,x,y,z,\alpha ,\beta ,\gamma \right]}^{T}$ representing the robot position (x,y,z) and orientation (α,β,γ) in the world frame is collected. Here, $t_{p},t_{pl},t_{in}$, and $t_{re}$ represent the time taken for each of the sub-tasks (pick, place, insert, and retract, respectively). Then, the GMM fitting is performed with K=7 components and the corresponding Gaussian regression allows one to compute the trajectory r(t), which is fed as the input to slave robot impedance control.

**Figure 9 sensors-23-08721-f009:**
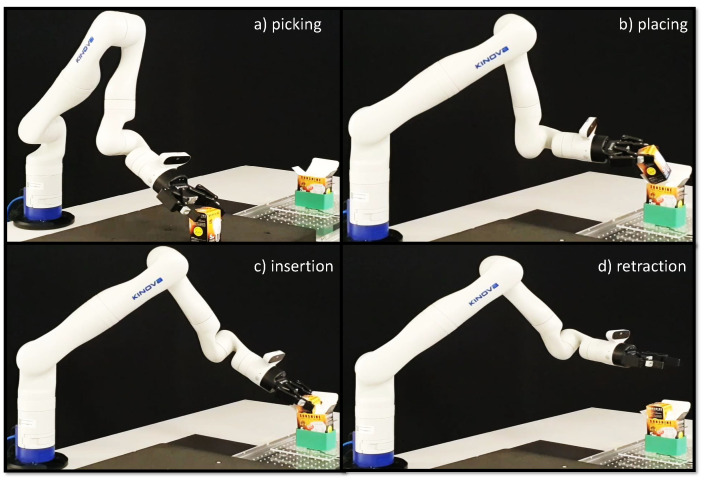
Automated insertion based on the demonstrated trials for a new box B position (video: refer to Appendix A).

## Data Availability

Not applicable.

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
