# Peer review of "Learning Compliant Box-in-Box Insertion through Haptic-Based Robotic Teleoperation"

_sensors, 2023, doi:10.3390/s23218721_

Round 1
Reviewer 1 Report
The current paper proposes a box-in-box insertion framework that uses learning from the demonstration paradigm in contrast to the conventional programming intensive approaches. The theory is validated using experiments.
Comments to authors:
- Please add more details of how the theory from the first sections is applied in the results section.
- Define and detail all the parameters used.
- Define all the acronyms used in the current paper.
- In the current version of the paper the novelty and contributions is not very clearly highlighted.
- In this paper the authors talk about optimal control, why no optimization function is specified.
- Add both the advantages and the disadvantages of the proposed method, in the current version of the paper only the advantages are presented.
- The state of the art it is poor regarding new habits that kids can acquire, maybe they can learn about automatic control, maybe the author could add the following publications:
o Hybrid Data-Driven Fuzzy Active Disturbance Rejection Control for Tower Crane Systems, European Journal of Control, vol. 58, pp. 373-387-11, 2021.
o Enhanced P-type Control: Indirect Adaptive Learning from Set-point Updates, IEEE Transactions on Automatic Control, DOI: 10.1109/TAC.2022.3154347, 2022.
- Please add more comments regarding the obtained results, and what initial conditions were used to obtain the current results?
Reviewer 2 Report
“Learning Compliant Box-in-Box Insertion through Haptic-based Robotic Teleoperation” by Kana et al. has developed a new framework for tight insertion by leveraging the natural compliance of objects, which are boxes, specifically in this work. Conventionally, the available works found in the literature focus on the compliance of the robot itself without utilizing the compliance of the objects. The authors have proposed and validated the approach utilizing the compliance of the objects and Learning from Demonstration techniques that incorporate three critical phases: demonstration, learning, and generalization. The overall approach could potentially make the autonomous insertion more tolerant of errors in localizing the pose of the boxes. The article is almost well formulated, and the results offer a timely engineering improvement to the solution of the box-in-box insertion problem. I have the following comments to further enhance the quality of this work.
1. The authors claim that the object’s compliance (a box made of deformable cardboard, specifically) could enhance the error tolerance of the autonomous insertion. Then, is there any quantified correlation between compliance and tolerance? Although the authors have demonstrated that for the specific object chosen here, the proposed method can be validated, it is worthwhile discussing the correlation for distinct compliance to further improve the impact of this work.
2. As a follow-up to the previous comment, could the promising approach also be validated on other deformable objects? Please discuss.
3. Is there any benchmark that authors have used to compare and further justify the superiority of the proposed approach? Please elaborate and discuss this.
4. In the Conclusion and Future Work, more directions of future explorations besides the effect of object orientation need to be supplemented.
5. The general field of object grasping, manipulation, and insertion is very exciting, and more directions are emerging due to the active advancement of soft robotics, control theory, and machine learning. Many interesting works that involve these topics should be cited here to further improve the impact:
rigid body grasping and manipulation: https://ieeexplore.ieee.org/document/9392354
Soft body grasping and manipulation: https://doi.org/10.1126/scirobotics.adg3792
Underwater grasping and manipulation: https://doi.org/10.1126/sciadv.adg0292
The overall English is good.
Round 2
Reviewer 1 Report
The authors improved their paper since the version and from my point of view the paper can be accepted as contribution in Sensors Journal.